# What are the implications of Zika Virus for infant feeding? A synthesis of qualitative evidence concerning Congenital Zika Syndrome (CZS) and comparable conditions

**Christopher Carroll**[1]*, **Andrew Booth**[1], **Fiona Campbell**[1], **Clare Relton**[2]

**1** Health Economics & Decision Science Section, School of Health and Related Research (ScHARR), University of Sheffield, Sheffield, United Kingdom, **2** Senior Lecturer in Clinical Trials, Institute of Population Health Sciences, Queen Mary University of London, London, United Kingdom

* C.Carroll@sheffield.ac.uk

**Data Availability Statement:** All relevant data are within the manuscript and its Supporting Information files.

## Abstract

If a mother contracts the Zika Virus before or during pregnancy, then there is a risk of the child developing Congenital Zika Syndrome (CZS). An infant can then experience problems feeding due to the specific physical and developmental consequences of Congenital Zika Syndrome (CZS), such as microcephaly, dysphagia and an increased likelihood of choking. This qualitative evidence synthesis accesses direct and indirect evidence to inform WHO infant feeding guidelines. We conducted a qualitative evidence synthesis of the values and preferences of relevant stakeholders (e.g. pregnant women, mothers, family members and health practitioners) concerning infant (0–2 years) feeding in the presence of: 1) CZS (the'direct evidence'); 2) severe disability and nonprogressive, chronic encephalopathies ('indirect evidence'), which present with similar problems. Authors' findings were extracted, synthesised using thematic synthesis techniques, and confidence in the findings were assessed using GRADE-CERQual. Six CZS-specific studies (all from Brazil) were included in the direct evidence, with a further eight indirect studies reporting feeding difficulties in infants with severe disability and nonprogressive, chronic encephalopathies. Included studies highlighted: breast-feeding represented the preference for all mothers in the studies in both reviews, and the inability to do so affected bonding between parents and child, and generated fear and anxiety relating to feeding choices, especially around the risks of choking and swallowing; the perception that health professionals were often unable to offer appropriate advice; the potential value of training; and a strong desire to achieve individual maternal autonomy in infant feeding decisions. Confidence in most findings ranged from low to moderate. The evidence base has limitations, but consistently reported that parents of children with feeding difficulties due to Congenital Zika Syndrome, or similar, need information, advice and counselling, and substantial emotional support. Parents perceive that these needs are often neither recognised nor satisfied; optimal feeding and support strategies for this population have not yet been identified.

**Funding:** This work was commissioned from the University of Sheffield, UK by the Department of Nutrition and Food Safety, World Health Organization, Switzerland as a technical document to support WHO recommendations on infant feeding. WHO grant 2019/887931-0 funded this research. The manuscript represents the views of the named authors only. The funders had no role in study design, data collection and analysis, decision to publish, or preparation of the manuscript.

**Competing interests:** The authors have declared that no competing interests exist.

## Author summary

If a mother contracts the Zika Virus during pregnancy, there is a risk of the child developing Congenital Zika Syndrome (CZS), which can lead to feeding difficulties due to swallowing and choking. It is therefore important to understand the lived experiences of those who support infants who have contracted CZS, so that infant feeding can be improved. The review and synthesis identified only a small number of relevant CZS qualitative studies (n = 6), all from Brazil (the 'direct' evidence). As a result, the review was expanded to include qualitative studies of the lived experience of mothers and others regarding infant feeding in the presence of similar physical problems, e.g. Cerebral Palsy (the 'indirect' evidence, n = 8). The phenomenon of interest was the same, though the underlying conditions were different. Both the direct and indirect evidence documented parental fear, anxiety and uncertainty around the problems they faced; the difficulties for maternal-child bonding; the potential value of training; and emphasised the perception that health professionals were poorly equipped to provide relevant information and advice. In particular, mothers wanted the specific feeding difficulties of their children to be taken into account when receiving advice and counselling. This synthesis has informed the WHO infant feeding guidelines.

## Introduction

Zika virus is a mosquito-borne virus transmitted by Aedes mosquitoes. The virus is prevalent in Africa, the Americas, Asia and the Pacific [1]. Infection with Zika virus usually results in mild illness. Symptoms may include fever, skin rashes, conjunctivitis, muscle and joint pain, malaise, and headache[2]. The Zika virus can be transmitted from mother to child in the womb, if the mother already has Zika or contracts the virus during pregnancy[3–5]. There is no current evidence of transmission via breast milk. A 2017 systematic review described three cases of Zika-infected breastfeeding mothers who were symptomatic within three days of delivery, and two cases with Zika-infected newborns[6]. The review concluded that, although Zika was detected in the breast milk of all three mothers, the data were not sufficient to conclude Zika transmission via breastfeeding. The authors stated that more evidence was needed to distinguish breastfeeding transmission from other perinatal transmission routes. In the light of insufficient reports of Zika virus being transmitted to infants through breastfeeding and, in countries with ongoing transmission of Zika virus, with no reports of adverse neurological outcomes in infants with postnatally acquired Zika virus disease, the World Health Organization (WHO) (2016) issued interim guidance on breastfeeding in the context of Zika virus[7]. They concluded that benefits of breastfeeding for the infant and mother outweigh any potential risk of Zika virus transmission through breast milk: infants born to mothers with suspected, probable or confirmed Zika infection, or who reside in or have travelled to areas of ongoing Zika transmission, should be fed according to normal infant feeding guidelines[8]. They should start breastfeeding within one hour of birth, be exclusively breastfed for six months and have timely introduction of adequate, safe and properly-fed complementary foods, while continuing breastfeeding up to two years of age or beyond [8].

In 2019, the WHO sought to update this guidance. In addition to commissioning a review update on the epidemiological evidence, they commissioned a Qualitative Evidence Synthesis (QES) to help to understand the values and preferences of pregnant women, mothers, their families, health care workers and service providers concerning infant feeding in the context of Zika infection. No published qualitative evidence had explored perceived risk from breast feeding, and related maternal preferences, in the context of the Zika virus. This lack might be

because the previous WHO Zika guidance had suggested that this risk was negligible, so studies have not been undertaken. However, as noted above, while the risk of transmission via breastmilk is negligible, there is a definite risk of transmission in utero, if the pregnant mother has or contracts the Zika virus, and increasingly, evidence does suggest that Zika virus has a causative role in neurological disorders[9]. Where a mother is infected within a narrow window of gestation a child may exhibit neurodevelopmental disorders that disrupt normal infant feeding. Although Zika virus disease is generally mild, increased cases of congenital microcephaly and Guillain-Barré syndrome have been observed in the Americas and the Pacific[8]. Congenital Zika Syndrome (CZS), including Zika-induced microcephaly, can affect an infant's ability to feed and swallow properly, either from the breast or when using cups or bottles[10]. Leal and colleagues report dysphagia in the first months of life of children with CZS[11]. Ten out of 13 infants in a case series of CZS without microcephaly at birth manifested dysphagia, including children with neurologic manifestations that were not very severe [12]. Feeding problems include lack of swallowing coordination; abnormalities of posture; and abnormalities of digestive tract motility, such as gastroparesis and gastroesophageal reflux[5]. Significantly, Leal and colleagues[11] point out similarities between Zika and swallowing difficulties associated with cerebral palsy[13]. They attribute CZS-associated dysphagia to anomalies of orofacial anatomy, oral and upper respiratory tract sensitivity, and changes in the motor function of the upper digestive tube caused primarily or secondarily by direct action of the virus[11]. Diverse conditions thus affect the ability of an infant with CZS to feed and swallow properly, either from the breast or when using cups or bottles.

The emphasis of the WHO-commissioned Qualitative Evidence Synthesis (QES) therefore shifted towards understanding the values and preferences of pregnant women, mothers, their families, health care workers and service providers concerning: infant feeding options (breastfeeding, use of breastmilk substitutes or mixed feeding with both breastmilk and breastmilk substitutes) when there are feeding issues on account of CZS. This evidence was directly relevant to the problem. The WHO then approved expansion of this qualitative evidence synthesis to the values and preferences of pregnant women, mothers, their families, health care workers and service providers concerning: infant feeding options (breastfeeding, use of breastmilk substitutes or mixed feeding with both breastmilk and breastmilk substitutes) when there are feeding difficulties in infants and toddlers with severe disability and nonprogressive, chronic encephalopathy as a result of conditions other than CZS. This indirect evidence was also considered by the WHO Guideline Committee to hold potential value in decision making. The previous 2016 guideline had not taken into consideration the special requirements for infant feeding among infants affected by Congenital Zika Syndrome (CZS), only whether there was a risk of transmission.

An earlier WHO document *Breastfeeding counselling: a training course*[14], had stated that mothers and families of infants born with congenital anomalies (e.g. microcephaly), or those presenting with feeding difficulties as in the case of an infant with CZS, should be supported to breastfeed their infants. Skilled feeding support from health professionals, including breastfeeding support, should be provided. Most Zika qualitative studies are predicated on breast feeding as the preferred option. Important contextual detail on infant feeding practices in the focal population is offered by a non-qualitative study on nutritional status and food practices in infants with Zika, at birth and 12–23 months of age[15]. Approximately 78% of infants were exclusively breastfed for less than six months. Before the first year of life, more than 90% of them were already consuming non-maternal milk. Difficulties with breastfeeding were reported by 53.6% of the mothers. These difficulties explain, at least partially, the high prevalence of early weaning: at twelve months less than 20% were continuously breastfeeding, whereas in children without microcephaly, this

frequency was approximately 35%. The authors state that the low prevalence of breastfeeding is to be expected given that children with microcephaly due to Zika virus exposure may present with dysphagia from the third month, when changes in oral-motor coordination, swallowing and sucking make breastfeeding challenging.

This QES therefore sought to explore the respective contributions of direct evidence (the CZS synthesis) and indirect evidence (the non-CZS related conditions synthesis) in understanding the values and preferences of pregnant women, mothers, family members and health practitioners, policy makers and providers (midwives) concerning feeding when infants experience physical problems with feeding and swallowing either due CZS or a condition/complex needs presented similar problems but unrelated to a transmissible disease. In this context, infant feeding and feeding support were defined as the alternatives to or most effective means of breastfeeding, if necessary, and the support required to facilitate feeding in such circumstances.

## Methods

We conducted a qualitative evidence synthesis in accordance with current best methodological practice and reported this according to PRISMA-derived ENTREQ guidelines[16]. We included studies that focus on those impacted by infant feeding challenges as a consequence of neurodevelopmental disorders as a result of: 1) CZS, when mothers contract the Zika virus before or during pregnancy; and 2) severe disability and nonprogressive, chronic encephalopathies other than CZS, e.g. Downs Syndrome.

### Reflexive note

In keeping with quality standards for rigour in qualitative research, the review authors considered how their views and opinions on infant feeding might influence decisions made in the design and conduct of the review. Furthermore, they considered how the emerging results of the study influenced those views and opinions. All authors believed, in line with WHO guidance, that breast feeding is the preferred method of infant feeding whenever possible, both on health grounds and, in low-and-middle income countries, for resource related reasons. All believed that positive infant feeding experiences are important for the wellbeing of the mother, baby, and the family, in the short and longer term. We therefore used refutational analytic techniques to minimise the risk that these prior beliefs would skew the analysis and the interpretation of the findings[17]. These techniques, initially outlined in the context of meta-ethnography, seek to create opportunities for identifying the "disconfirming case" and include introducing different levels of familiarity of the data and disciplinary perspectives and challenging hierarchical and power relations within the research team[17].

The principal synthesis of direct evidence constitutes a qualitative evidence synthesis of a specific subset of studies relating to the Zika Virus identified during the course of a larger QES on values and preferences related to infant feeding in the presence of transmissible infection. This main Zika synthesis, prioritised and performed to the standards of a conventional QES, initially combined the concepts of Zika Virus, Breastfeeding, Infant feeding and Qualitative Research. However, a supplementary literature search with extreme sensitivity (Zika Virus and any Qualitative Research) allowed identification of (i) data where keywords or indexing had not originally identified a study as potentially relevant; and, (2) data where breastfeeding considerations are mentioned in passim or alongside other issues.

The need to include indirect evidence from infants with severe disability and nonprogressive, chronic encephalopathy had been identified by the guideline panel given a paucity of data from direct studies of the Zika virus. Behaviourally and practically mothers contend with these closely-related phenomena of interest. Importantly, Leal et al observe that "few

articles have been published on dysphagia in children with severe disability and nonprogressive, chronic encephalopathy; most published articles refer to dysphagia in persons with cerebral palsy"[11]. Dysphagia in children with different neurologic aetiologies is "characterized by considerable variability"[11]. GRADE-CERQual guidance on relevance[18] acknowledges the potential value of similarly-related "indirect evidence" in informing the Zika-guidelines on infant feeding. The principal conditions of potential relevance as indirect evidence, in which infant feeding difficulties comparable to CZS are present, include the likes of Cerebral Palsy, Down Syndrome, Guillain Barre Syndrome, Pierre Robin Syndrome and STORCH (syphilis, toxoplasmosis, rubella, cytomegalovirus or herpes simplex). The synthesis of studies of infants with severe disability or nonprogressive, chronic encephalopathy was conceived within the time pressures of a rapid QES framework; an emerging methodology that currently lacks agreed standards for conduct[19]. Addition of indirect evidence on infants with severe disability and nonprogressive, chronic encephalopathy admitted studies from populations located in non-Zika geographical areas.

## Protocol and registration

The CZS review was originally conceived as a subset of studies within a broader review of *Acceptable medical reasons for use of breast-milk substitutes in the context of transmissible disease*. Therefore, no separately published protocol exists for the synthesis of CZS studies. The secondary synthesis, providing indirect evidence on infants with severe disability and nonprogressive, chronic encephalopathy, was subsequently conducted as a rapid synthesis. Consequently, both syntheses (Table 1) share their methods with those for the parent *Acceptable medical reasons for use of breast-milk substitutes* review)[20]. PROSPERO 2019 CRD42019143387.

Table 1. Overview of the evidence syntheses.

| | Conventional Qualitative Evidence Synthesis of Infant feeding in the Context of Congenital Zika Syndrome (CZS) | Rapid Qualitative Evidence Synthesis of Infant feeding in the Context of Severe Disability and Nonprogressive, Chronic Encephalopathies |
|---|---|---|
| Publication Dates | 2000–2019 | 2000–2019 |
| Perspectives | Women, partners, carers and significant others, healthcare providers, policy makers | Women, partners, carers and significant others, healthcare providers, policy makers |
| Qualitative Outcomes of Interest | Values and Preferences | Values and Preferences |
| Study types | Qualitative Research Studies (Quantitative research studies including qualitative outcomes were excluded] | |
| Search Methods Used | Subject searching, reference checking, citation searching. Filter for qualitative research. | Subject searching, reference checking, citation searching. Abbreviated filter for qualitative research. |
| Sources Used | • CINAHL (Ovid)<br>• MEDLINE (Ovid)<br>• EMBASE<br>• PsycINFO (Ovid);<br>• Social Science Citation Index (Web of Science); SCIELO, Scopus, POPLINE, LILACS, BIREME<br>• African Journals Online<br>• Google Scholar | • PubMed<br>• CINAHL<br>• Web of Science<br>• SCIELO,<br>• Scopus<br>• LILACS<br>• BIREME<br>• African Journals Online<br>• African Index Medicus<br>• Google Scholar |
| Method of Synthesis Used | Thematic Synthesis | Thematic Synthesis |
| Relevance* | Direct | Indirect |

* Relevance[18] is the GRADE-CERQual component that corresponds most closely to the GRADE component of Indirectness[21]

Our experience in conducting the rapid qualitative evidence synthesis mirrors that reported by Downe and colleagues in the context of QES to support World Health Organization Guideline development. They describe the need for "'mini-reviews' of the qualitative evidence in specific areas (in our case of swallowing difficulties) . . .undertaken as the guideline development process progressed"[22]. By definition these mini-reviews utilise more focused searches and more limited choice of sources, to identify "indirect evidence" when compared to the core QES review (Table 1).

## Focused questions and inclusion criteria

The focused questions for the two, complementary reviews were as follows:

- Direct evidence (CZS): What are the values and preferences of pregnant women, mothers, family members and health practitioners, policy makers and providers (midwives) concerning feeding when an infant has difficulties as a result of CZS?

- Indirect evidence (similar but non-CZS related feeding problems): What are the values and preferences of pregnant women, mothers, family members and health practitioners, policy makers and providers (midwives) concerning feeding when infants experience problems with feeding and swallowing due to a condition or complex needs unrelated to transmissible disease, such as Zika?

To be included in the review and synthesis, studies were required to satisfy criteria, defined using the PerSPEcTiF(S) framework[23] (Table 2).

**Information sources and search strategies.** Literature searches were conducted between April and November 2019 using four main search strategies:

1. Broad search for qualitative research on infant feeding for Zika (primary studies) (April 2019).

2. Follow up of included reviews and primary studies, checking references and citation searching of all included references on infant feeding for Zika (May-July 2019).

3. Broad search for any qualitative research on Zika Virus (NB. Dropping explicit requirement for infant feeding in title/abstracts) (August 2019).

4. Rapid targeted searches on infant feeding in infants with severe disability or nonprogressive, chronic encephalopathies (e.g. microencephaly, dysphagia, choking, swallowing etc) (September 2019) and on specific conditions e.g. cerebral palsy and Down's Syndrome (October 2019).

**Table 2. Inclusion criteria, defined using the PerSPEcTiF(S) framework[23].**

| Infant feeding in the context of: | Congenital Zika Syndrome (CZS) | Severe disability and nonprogressive, chronic encephalopathies |
|---|---|---|
| Perspective(s) | Women, partners, carers and significant others, healthcare providers, policy makers | |
| Setting: | Any setting (primarily community settings) | |
| Phenomenon of interest: | Infant feeding in the context of CZS | Infant feeding in the context of difficulties in feeding due to severe disability and nonprogressive, chronic encephalopathies |
| Environment: | International, particularly Low- and Middle-Income countries (LMICs) where the Zika Virus is prevalent | |
| Comparison | (Implicitly compared with other parents with infants experiencing feeding difficulties] | |
| Timing: | When contemplating, carrying out or supporting breastfeeding, breast milk feeding or alternative infant feeding | |
| Findings: | Fears, perceptions, experiences, beliefs values and preferences regarding the phenomenon of interest | |
| Study Design | Qualitative studies. Surveys with qualitative data as free text responses to survey questions were excluded | |

Search methods are described sequentially according to these phases. We searched the following databases for relevant published and unpublished literature from 2000 to 2019: PubMed; MEDLINE (Ovid); PsycInfo (Ovid); CINAHL (Ovid); EMBASE (Ovid); Web of Science; SCIELO; Scopus; LILACS (for studies conducted in South America); BIREME; African Journals Online (for studies conducted in Africa); and African Index Medicus. Using guidelines developed by the Cochrane Qualitative & Implementation Methods Group for searching for qualitative evidence[24,25] search strategies were developed for each database. The search combined thesaurus and free-text terms for Zika with terms for infant feeding and published filters to identify qualitative research (https://hiru.mcmaster.ca/hiru/HIRU_Hedges_MEDLINE_Strategies.aspx). Subsequently, the search was broadened using a search tactic known as "drop a concept"[26]; in other words, the most specific concept, infant feeding, was omitted from the search strategy to retrieve qualitative research on Zika, regardless of whether or not the specific aspect of infant feeding was specified (S1 Table). This revised strategy combined thesaurus and free-text terms for Zika with an abbreviated filter to identify qualitative research[24,27]. The search strategy for MEDLINE is available in the S2 Table. Reference lists of included papers were scrutinized (back-chaining) and items included as appropriate. Citation searches (forward-chaining) were run on Google Scholar for all included studies using the Publish or Perish software[24]. Google Scholar alerts were set up for all included studies to identify newly published studies published up until the final analyses, and any further studies retrieved for inclusion. Results were transferred to Microsoft Excel spreadsheets for sifting for possible inclusion.

Non-English language studies were included where translation was possible. Titles and/or abstracts of potentially relevant studies published in languages other than English were translated using a basic translation package (Google Translate). No geographic restrictions were imposed on the search; the date range was limited to 2000–2019 to capture recent and contemporary views.

## Study selection, data extraction and quality assessment

AB and CC each independently screened the titles and abstracts of all of the initial hits against the inclusion criteria, referring queries for agreement by consensus. In the event an article could not be categorically excluded, the full paper was retrieved. AB and CC then each independently screened the full texts of all identified papers against the inclusion criteria, referring queries for agreement by consensus. Included studies were subject to quality assessment using the Critical Appraisal Skills Programme checklist for qualitative research[14], modified to exclude questions relating to applicability. This simple appraisal system rates studies against 10 criteria with a focus on the appropriateness of the study design, methods of data collection and data analysis. While not validated to detect risks to rigour, this tool is the most commonly-used instrument for qualitative research and offers a structured and consistent approach to study quality assessment. No studies were excluded on the basis of study quality; even studies with design flaws may offer valuable insights within an interpretive framework[28].

## Analytic strategy

Thematic synthesis techniques were used for analysis and synthesis, with data being summarised on a 'significant extracts' basis; that is to say, by using full or substantive verbatim quotations from participants rather than by line-by-line coding. In step one, the included papers offering direct evidence (the Zika studies) were examined, and an index paper was selected, chosen on the basis of richness. Themes and findings identified by the authors of this paper were entered onto a spreadsheet, to develop an initial thematic framework. The findings of all remaining papers were then mapped to this framework, with unique findings used to generate

new themes where relevant. If disconfirming data were found, themes were amended, to capture all the data from papers already analyzed, as well as taking account of new insights. This process ensured that the final analysis held high explanatory power for all the data. This process was repeated for indirect evidence studies. Finally, integration was undertaken by examining similarities of themes, as well as nuanced differences across the two literature sets.

Themes were agreed by consensus between CC and AB, and subject to scrutiny by FC and CR. All major findings were supported by extracted text and/or verbatim quotations from more than one included study. All findings were assessed for confidence in the quality (methodological limitations), coherence, relevance and adequacy of the contributing data, using the GRADE-CERQual tool[29]. This approach mirrors the Grading of Recommendations, Assessment, Development and Evaluation (GRADE) approach used in effectiveness reviews. The GRADE-CERQual assessment results in a final classification of confidence in the theme in four categories: 'high', 'moderate', 'low' or 'very low'.

## Results

### Included studies

A total of 14 qualitative studies (six CZS and eight non-CZS) were included across the two syntheses.

### Direct evidence: Congenital Zika Syndrome (CZS) review

The primary search strategy for the CZS synthesis generated 778 potentially relevant records (Fig 1). One further reference was identified from reference lists. 296 duplicate studies were removed, leaving 483 to be screened. 431 records were excluded at title and abstract stage as unrelated to the topic of interest. The remaining 52 were processed at full text review. A further 46 articles were excluded at the full text stage leaving six CZS studies (see Fig 1).

Six publications satisfied the inclusion criteria (see Table 3). All studies were conducted in Brazil and focussed on the values and preferences of parents or carers of children ith CZS. Four studies did not specify the age of the affected children, but it was inferred that these satisfied the inclusion criteria because of the references to 'infants', parental experience of swallowing difficulties in the first days and weeks, and the use of health services for diagnosis and management of microcephaly. Two studies focused on parents and family (n = 23), principally mothers, either using focus groups to collect data,[30] or analysing parents' YouTube videos depicting their experience[31]. Two studies, one focusing on mothers of children with Zika-related microcephaly (n = 12)[32] and one on fathers of these children (n = 5)[33], employed interviews alone. An ethnographic study collected data from field-notes of researchers who followed 'caretakers' (parents) of children with Zika-related microcephaly as they sought information, support and service provision[34]. Finally, one study examined the experiences of 70 mothers as they underwent an educational intervention aimed to improve the local health promotion response to microcephaly.

### Synthesis

**Parental anxiety and stress.** *Parents report uncertainty about how best to feed their child due to swallowing difficulties*. In an analysis of YouTube videos posted by relevant family members, it was recorded that families first approached health professionals only 'after a few days with their child, [when] mothers recognized abnormal infant behaviors, such as constant crying and difficulty in swallowing'[31]. Parents in one CZS study reported a need for guidance on infant feeding, especially on the management of breastfeeding, weaning, types of foods,

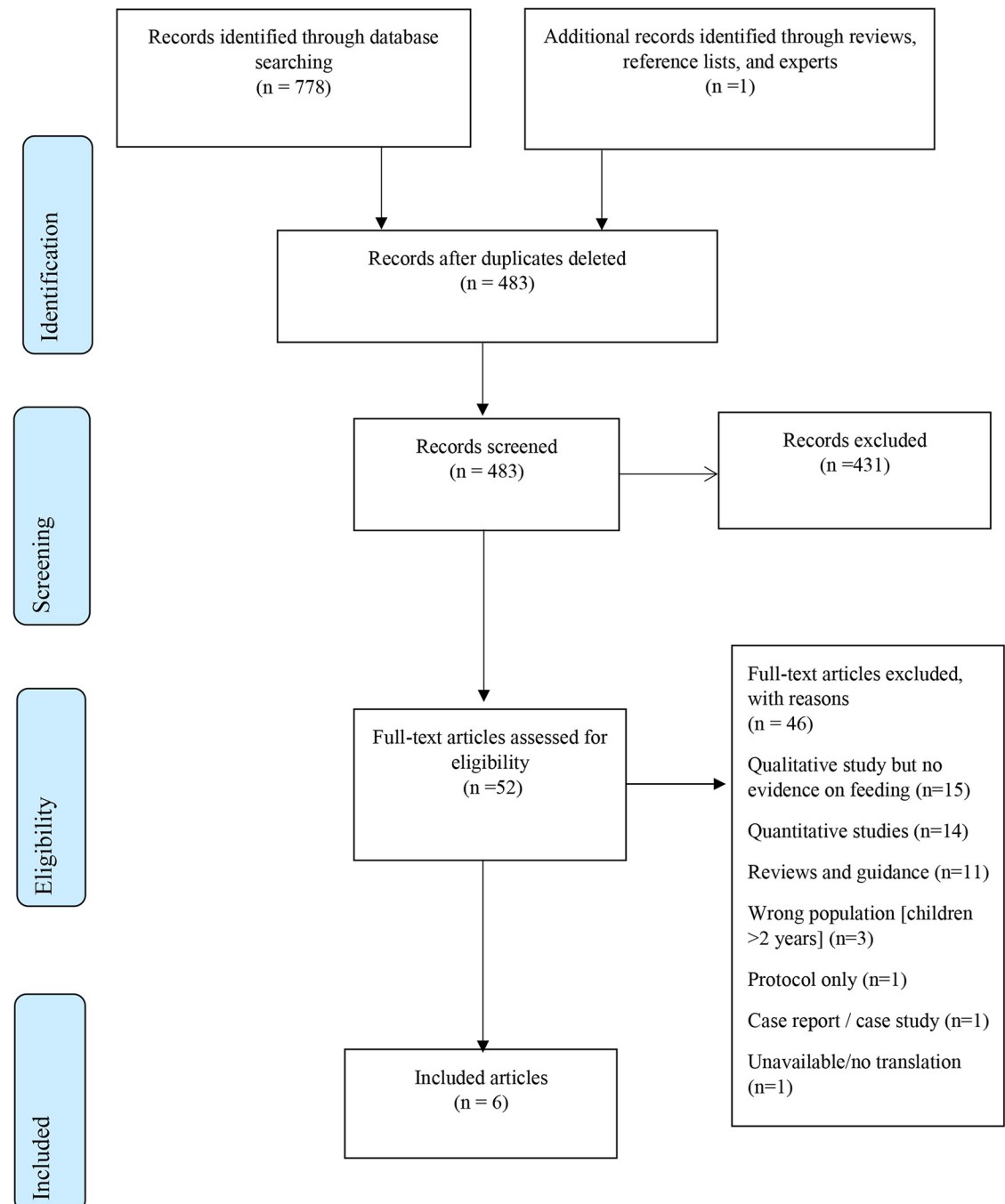

**Fig 1. PRISMA Flow Diagram for review of CZS qualitative studies.**

spacing of diets, introduction of supplements and use of adequate pacifiers when the baby is not exclusively breastfed [30]. Participants in three studies reported that one of their concerns was that they did not know how best to feed their child[30,33,35]. Clear uncertainty and 'doubt' existed regarding the best diet and the best way to feed their child. Parents reported a lack of knowledge regarding how to manage choking or gagging[30,33]; parental uncertainties

**Table 3. Congenital Zika Syndrome (CZS) review—Characteristics of included studies.**

| Author (Date) | Setting (i.e. Country) | Vicinity (i.e. Region, State, Province, City) | Study Aims and Purpose | Conditions Included | Perspectives and sample characteristics | How was the sample selected? | Data collection methods used? |
|---|---|---|---|---|---|---|---|
| De Sa (2017)[30] | Brazil | Fortaleza, Ceará | To identify parental needs with respect to the care for the development of infants and children with microcephaly caused by the Zika virus (ZIKA). | CZS | Parents and families of infants with zika-induced microcephaly (n = 23). Children's age: unspecified | Convenience | Focus groups |
| Da Silva Rodrigues Felix (2018) [33] | Brazil | North-east | To investigate the impact of the birth of infants with microcephaly on the family dynamics, based on the father's perceptions. | CZS | Fathers of infants with zika-induced microcephaly (n = 5). Children's age: unspecified | Purposive | Interviews |
| Campos (2018)[32] | Brazil | Fortaleza, Ceará, | To understand the challenges and perspectives of mothers of children with microcephaly due to Zika virus infection. | CZS | Mothers of infants with zika-induced microcephaly (n = 12). Children's age: 12–26 months | Convenience | Interviews |
| Scott (2018) [34] | Brazil | Recife | To understand the way different contexts (discovery, household, health units, social work, associations) contribute to creating notions about maternity and childhood. | CZS | Researchers (n = 19) working with 'caretakers' of children with microcephaly. Children's age: unspecified. | Convenience | Ethnography fieldwork / observation |
| Dos Santos (2019)[35] | Brazil | 3 reference centers for treatment of children with microcephaly in the state of Sergipe (North east) | To report educational experiences of mothers or caregivers of children with microcephaly, with a view to development by academic staff of health promotion for these children. | CZS and STORCH (syphilis, toxoplasmosis, rubella, cytomegalovirus or herpes simplex) | Mothers or primary caregivers of children with a confirmed diagnosis of microcephaly (n = 70). Children's (mean) age: 15 months | Convenience | 1) Participant observation with field diaries and 2) interview with mothers through unstructured script of questions |
| Vale (2019) [31] | Brazil | North-east | To understand the experience and perceptions of families with infants diagnosed with Zika-related microcephaly | CZS | Parents and 'family': mothers (n = 27), fathers (n = 6), grandmother (n = 1), nanny (n = 1). Children's age: unspecified. | Convenience | Youtube videos |

CZS: Congenital Zika SyndromeThe evidence base was at low / moderate risk of bias (see S1 Fig). All but one study[34] presented a clear question; all studies had a qualitative design and used appropriate methodology, but with a moderate risk of bias in reported recruitment, data collection and analysis strategies in some studies. Two studies did not report clear findings[31,34]. However, these two studies were the only ones to address the relationships between researchers and participants (reflexivity)[31,34]. Illustrative quotations supporting some of the themes below are presented in S3 Table: Themes and illustrative quotations: Feeding infants with Congenital Zika Syndrome.

that were reflected in more aspects that feeding alone [36]. Participants in a further study received an educational intervention that covered choking problems[35].

*Parents report fear and anxiety due to choking or difficulty swallowing.* The theme of choking or difficulty swallowing appeared in four Zika studies[30,33]. Participants in two studies reported that their principal concern was their child choking[30,33]. Parents in these studies also reported that they were unsure how to manage choking or gagging [30,33]. Mothers in particular reported that feeding almost always produced choking [32]. Several parents reported in one study that, such was their level of fear and anxiety when it came to feeding their child, because the child

would choke, that they simply did not want to do so; they would rather do anything else[30]. Participants in two studies also reported a fear of suffocating their child when feeding[30,33].

Researchers in an ethnographic study reported that anxiety over choking could affect other female family and community members also: 'Other women are notably more reluctant to participate in holding these babies, because they .., have problems in swallowing food, suffer frequent convulsions, and, in general, present a very vulnerable condition'[34]. Fifty percent of CZS babies have been reported to have poor lipid intake, and this inadequate dietary intake of lipids may be explained by the main challenges posed by choking, when mothers thicken food through the use of carbohydrate-rich farinaceous food[15].

*Bonding concerns when feeding is problematic.* Demands on the mother were seen to impair the relationship between mothers and their children. Mothers reported "not accepting the congenital anomaly that their child has and how it ends up reflecting on their daily habits and consequently on their bond with the child"[35].

*Parents report that the burden of feeding can be time-consuming and stressful for mothers.* A mother in one study reported that feeding a child with swallowing difficulties placed a time burden, as well as an emotional burden, on the primary carer, principally the mother [32].

## Health professionals

**Information and advice.**   One CZS study documented a mother's anecdotal report of her daughter's food related broncho-aspiration pneumonia acquired due to poor positioning during feeding, as she felt the instructions she had received werenot adequate [35]. Mothers in a further study reported how doctors, and other health professionals providing home visits, did offer guidance and instruction about food, but that this might be ignored: a mother reported that she was told not feed her child certain things, but she fed them and her son ate and enjoyed them anyway [32]. The perceived adequacy and value of information provided to parents appeared to differ in these studies.

## Support

**Training programmes for parents.**   In two CZS studies, mothers reported the information they needed from training programmes. Participants in a tailored training programme reported that they had difficulties with breastfeeding and introducing first foods, and that they needed explanations about the feeding positions they used with their babies and which foods they had introduced first[35]. Elsewhere families expressed a specific need to learn first-aid techniques, mainly "in order to reverse gagging, caused by dysphagia, and airway cleansing and hygiene maneuvers in children, since infants have gastric reflux and difficulty accepting food [especially the milk], making it difficult to feed them properly"[30]. Many mothers of children with microcephaly expressed doubts related to their children's diet, discussing such issues as the correct mode of food preparation and the types of foods introduced first, according to each child's age group[35]. The training programmes advised mothers about the importance of exclusive breastfeeding during the first 6 months, as well as stimulating the baby's grip, as well as providing advice on essential care when administering food by nasogastric and nasoenteral tubes [32]. No study mentioned any potential role for bottle-feeding.

**Resource considerations.**   Caring for a child with feeding problems places pressure on family resources. Researchers in an ethnographic study noted that, 'For many reasons the lack of resources becomes more intense for these families, especially when mothers leave their jobs and have growing needs for medicines, expensive special food, therapeutic instruments (to favor deglutation, ambulation, posture, vision, etc.) . . .')[34]. In a further study, mothers who were unable to maintain exclusive breastfeeding were instructed to purchase formula feed

 

offered by the state[35]. Lack of support from the government was highlighted by parents as the main cause of mothers' hopelessness towards feeding their children.

## GRADE-CERQual summary

Confidence in the findings regarding parental uncertainty on how best to feed their child was moderate, but confidence in all other findings was categorised as low or very low (Table 4). Studies used different methods of data collection and included observation of parents and family, as well as approaches that directly elicited their values and preferences, and were all published between 2017 and 2019, so provide contemporary evidence. However, the sample only included six studies from a single geographical region (north-east Brazil), which principally explored the views and experiences of mothers and fathers. The focus of the six included studies was not infant feeding, but rather parental responses to, and management of the general impact on parents and family of having a child with CZS. This was the principal focus of the wider literature also: mention of infant feeding was rare, and limited to these six studies. As a result, while these findings on key individuals' values and preferences for infant feeding are important, caution must be exercised because the population sample is highly-localised and the external validity of included studies (GRADE-CERQual–relevance) may therefore be limited.

**Table 4. CZS review—GRADE CERQual summary of findings for feeding infants with CZS.**

| Summary of review finding | Studies contributing to review finding | GRADE-CERQual assessment of confidence in the evidence | Explanation of GRADE-CERQual assessment |
|---|---|---|---|
| Parents report that they often do not know how best to feed their child with microcephaly because the child frequently chokes and has difficulty swallowing | [30–35] | Moderate confidence | Six studies with minor or moderate concerns about coherence, adequacy, relevance and methodological limitations (only five studies, but all are recent and conducted in the region in Brazil that is the principal location of CZS). |
| Parents and others report that feeding a child with swallowing difficulties makes them stressed and anxious, even if they possess information on how to manage this: they fear that they might be doing or do something wrong, and that they might suffocate the child | [30,33–35] | Low confidence | Four studies with minor concerns about coherence because the link between the data and findings is very clear, but moderate concerns about methodological limitations, adequacy and relevance (all three studies are recent and conducted in the one region in Brazil that is the principal location of CZS). |
| Mothers report that problems with feeding can affect bonding with their child | [35] | Very Low confidence | One study with serious or moderate concerns about methodological limitations, coherence, adequacy and relevance. |
| Mothers report that the burden of feeding, which can be time-consuming and stressful, falls on them | [32] | Very Low confidence | One study with serious or moderate concerns about methodological limitations, coherence, adequacy and relevance. |
| Parents feel that the information provided to them by health professionals is mostly inadequate | [32,35] | Very Low confidence | Two studies with serious or moderate concerns about methodological limitations, coherence, adequacy and relevance. |
| Families value training where it is given | [34,35] | Very Low confidence | Two studies with serious or moderate concerns about methodological limitations, coherence, adequacy and relevance. |
| Families experience economic pressures because of the need to buy special food | [34,35] | Very Low confidence | Two studies with serious or moderate concerns about methodological limitations, coherence, adequacy and relevance. |

CZS: Congenital Zika Syndrome. The full GRADE-CERQual Evidence profile table is available in S4 Table: GRADE-CERQual Evidence profile: Feeding infants with Congenital Zika Syndrome.

 

### Indirect evidence: Non-CZS review

Details of the study selection process are presented in Fig 2. The search retrieved 473 potentially relevant records. 443 records were excluded at title and abstract stage, and a further 22 articles were excluded at the full text stage.

Eight additional publications satisfied the inclusion criteria for indirect evidence on infant feeding/swallowing difficulties (see Table 5). One study was conducted in each of Bangladesh, Brazil, Portugal, and the UK with two each from Australia and Ghana. Three studies focused exclusively on mothers of infants with Down's Syndrome; these studies used either interviews

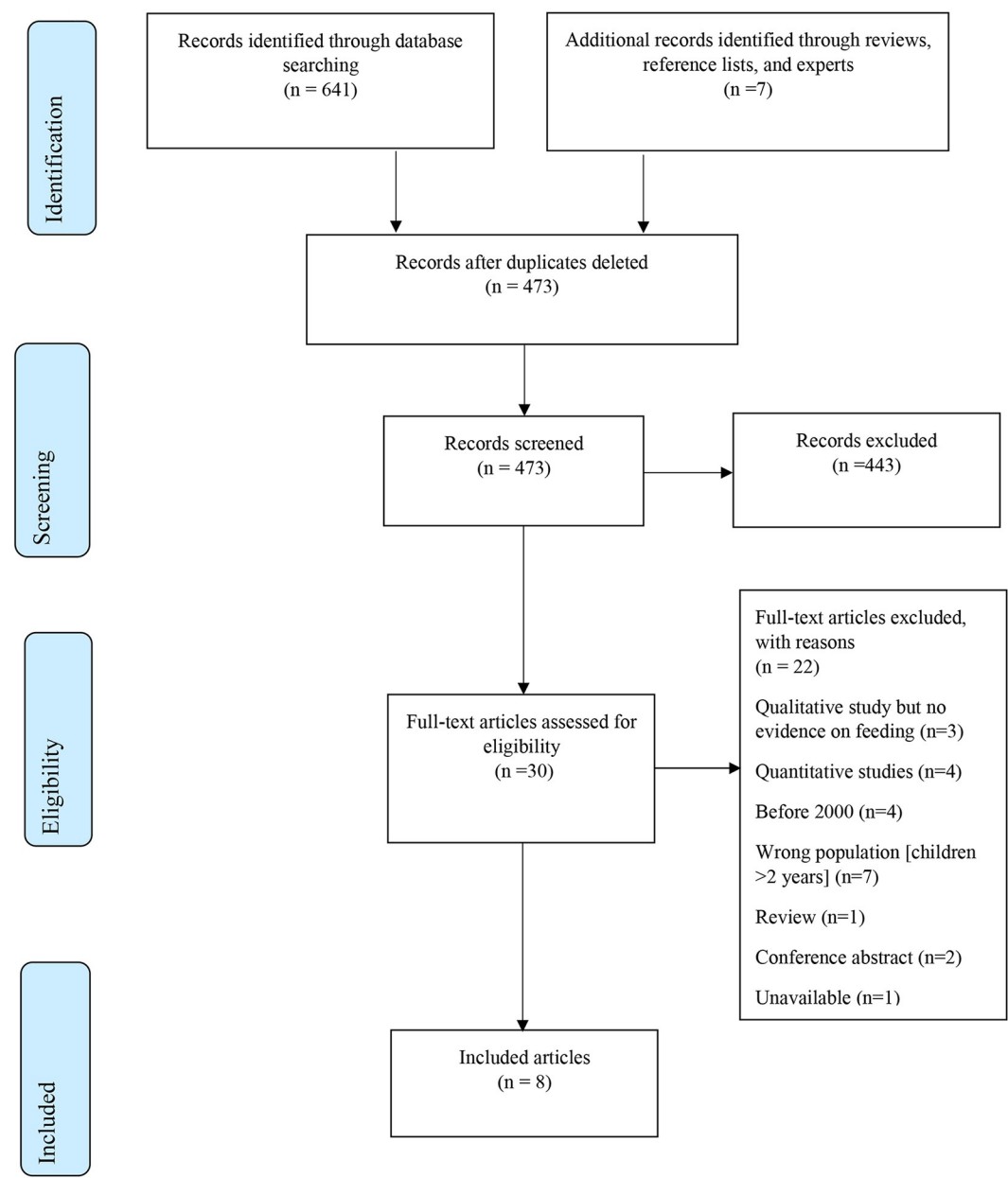

**Fig 2. PRISMA Flow Diagram for review of qualitative studies on severe disability and nonprogressive, chronic encephalopathy.**

**Table 5. Non-CZS review—Characteristics of included studies.**

| Author (Date) | Setting (i.e. Country) | Vicinity (i.e. Region, State, Province, City) | Study aims and purpose | Which conditions are included? | Perspectives and sample characteristics | How was the Sample selected? | Data collection methods used? |
|---|---|---|---|---|---|---|---|
| Adams (2011) [41] | Bangladesh | Dhaka | To design, implement and evaluate a low-cost intervention, to address the feeding difficulties of children with moderate–severe CP to inform appropriate service development for this population and their caregivers | Cerebral Palsy | 22 child–caregiver pairs. Children's age: 19–129 months | Opportunistic | Interviews |
| Barros da Silva (2018)[37] | Portugal | Porto | To understand the breastfeeding experiences of mothers of children with Down's Syndrome, including their perceptions of the breastfeeding process and their specific practices. | Down's Syndrome | Mothers (n = 10) of children with DS. Children's age: 2 mths– 9 yrs. | Snowball | In-depth semi-structured interviews |
| Cartwright (2018)[39] | UK | West Midlands, North-west | To explore the experiences of mothers of infants with Down's Syndrome regarding feeding, and to provide information to better inform health professionals caring for new mothers and their babies. | Down's Syndrome | Mothers (n = 8) with infants with Down's Syndrome. Children's mean age: 18 months (range 8 weeks—3 years) | Convenience | Focus groups |
| Donkor (2019) [42] | Ghana | Four geographical regions: Agogo, Dodowa, Sunyani, and Techiman | To explore caregiver experiences of feeding before and after a community-based training program in Ghana | Cerebral Palsy | 13 caregivers of severely undernourished children with CP at start of training program. 11 interviewed again after year of training and home visits.Children's age: 1 yr 5 mths—11 yrs 11 mths | Convenience | Interviews |
| Morrow (2007) [44] | Australia | Sydney | To identify the major determinants of feeding-related quality of life (QoL) in children with quadriplegic cerebral palsy (QCP) from the perspective of health professionals to provide a framework for comprehensive clinical evaluation of health status in this group | Cerebral Palsy | General and specialist paediatricians (n = 18), nurses (n = 15); allied health professionals (n = 13). | Purposive—to ensure variability in speciality, gender, years of experience and place of work | Five semi-structured focus groups |

(*Continued*)

**Table 5.** (Continued)

| Author (Date) | Setting (i.e. Country) | Vicinity (i.e. Region, State, Province, City) | Study aims and purpose | Which conditions are included? | Perspectives and sample characteristics | How was the Sample selected? | Data collection methods used? |
|---|---|---|---|---|---|---|---|
| Swift (2010)[40] | Australia | South Australia | To describe the experiences of both mothers and fathers with a child in a neonatal unit with a feeding difficulty at 36 weeks gestational age, with particular attention on the role of the feeding difficulty, the meaning of the experience for parents and the influence of and impact on, family relationships; second, to develop exploratory model from the data to identify considerations for current and future neonatal unit staff to be incorporated into daily practice to improve family-centred care | Difficulty latching on, fatigue while feeding, swallowing / breathing difficulties while feeding conditions unspecified | Mothers (n = 7) and fathers (n = 2).Children's age: 3–6 months old at time of interview | Purposive | Interviews and field notes |
| Wieczorkievicz & De Souza (2009) [38] | Brazil | Rio Negro-PR and Mafra-SC (South) | To describe the facilities found by women / mothers of children with Down's Syndrome in process of breastfeeding | Down's Syndrome | Mothers of children with Down's Syndrome (n = 6). Children's age: Unspecified | Selected from registration sheets obtained from Association of Parents and Friends of Exceptional Children (APAE). | In-depth interview |
| Zuurmond (2018)[43] | Ghana | Four geographical regions | To explore the impact of a participatory training programme for caregivers delivered through a local support group, with a focus on understanding caregiver wellbeing. | Cerebral Palsy | 18 primary caregivers; 14 mothers, 3 grandmothers, and one male cousin. Children's age: 18 mths—12 yrs | Purposively sampled | Interviews |

(n = 10 [37]; n = 6 [38]), or focus group participants (n = 8) [39] to collect data. One study focused on mothers and fathers [n = 9] whose babies had been in a Neonatal Intensive Care Unit, and who experienced one or more feeding difficulties[40]. This study used in-depth interviews and field-notes to collect data[40]. Four studies examined children with cerebral palsy; three were focused on family caregivers, primarily mothers[41–43], and one on health professionals[44].

In two studies, the mean age or age range of the children was reported as being less than 2 years and the parents reported experiences of different approaches to infant feeding[39,40]. In four studies the age range was partially overlapping with the focal ages for this review. In one study the age range of the children was 2 months to 9 years[37], in others it was 19–129 months[41], 1 year 5 months—11 years 11 months[42], and 18 months– 12 years[43]. In the last family caregiver study the age of the children was not specified[38]. Two studies focused on breastfeeding alone, suggesting that reported parental experiences related principally to children <2 years of age[37,38]. The study of health professionals did not specify an age group but it was assumed that they would have acquired wide experience across diverse age groups

[44]. Two studies were considered of moderate quality[41, 43] with the remainder exhibiting low risks to rigour. The data were relevant and rich. Illustrative quotations supporting all of the themes below are presented in S5 Table: Themes and illustrative quotations: Feeding in infants with severe disability or nonprogressive, chronic encephalopathies.

The evidence base was at low / moderate risk of bias (S2 Fig). All but one study[43] presented a clear question; all studies had a qualitative design and used appropriate methodology, but there was a moderate risk of bias in reported recruitment across six of the eight studies, as well as serious concerns about data collection and moderate concerns about data analysis in one study [40]. Finally, there were serious or moderate concerns regarding the relationships between researchers and participants (reflexivity) in five studies [37,38,40,41,43].

### Synthesis

**Parental anxiety and stress.**   *Parents report uncertainty about how best to feed their child.* In the UK and Australian studies, mothers consistently reported uncertainty about whether they were doing the best for their child, making the best feeding choices [39,40]. Mothers of UK Downs Syndrome infants reported that their infants were often sleepy and did not demonstrate feeding cues, which is not reported in the literature regarding typical infants, and they struggled to gain answers from health professionals [39].

*Bonding concerns when feeding is problematic.* Mothers in an Australian study reported that they felt that the requirements or advice to adopt a certain approach to feeding, such as expressing breast milk for bottle feeding, affected their ability to bond with their child [40]. However, Australian health professionals expressed the view that prolonged times spent feeding presented an important opportunity for one-on-one contact between the main carer and the child[44]. The study from Portugal found that family and professional support could help mothers to manage their different feelings and to enhance their positive feelings and thus help in the maintenance of breastfeeding, without early weaning[37]. Mothers in a study from Brazil felt that the professionals who attend the woman/mother of a child with Down syndrome should promote early bonding, if possible, taking the first hour to establish breastfeeding, because at this moment, despite likely drowsiness, the baby is likely to be active[38]. In this study, one mother also reported that prior familiarity with Down's Syndrome created a "bridge" to bonding, thereby increasing the chances of successful breastfeeding[38]. Health professionals in an Australian study felt that the shared responsibility of caring for a child with cerebral palsy might enhance family bonding[44]. Effects of caregiver feelings of self-stigma and shame were profound and were given as a major reason why fathers had left, family members were unwilling to care for the child, the child and caregiver faced exclusion in home and community life, and why the child may be neglected. This finding for cerebral palsy in Ghana[43] had previously surfaced in general studies of the effects of Zika in Brazil[45].

*Parents report the burden of feeding can be time-consuming and stressful for mothers.* Typically, studies reported that the feeding of an infant with feeding difficulties was time consuming[38–40,42,43,44]. Parents reported experiencing frustration that they could not feed in the way they wanted to, such as breast-feeding[40], and that interventions and advice often did not 'work'[39,40]. Mothers in the UK study of infants with Down's syndrome, reported finding feeding, whether breastfeeding, expressing or bottle-feeding, to be 'overwhelming', 'lengthy and time-consuming', 'exhausting' and 'difficult'[39]. This was echoed in an Australian study in which mothers reported finding expressing and feeding 'degrading', 'time-consuming' and leaving them 'feeling fatigued'[40].

Researchers in a Ghanaian study reported that long meal times were the norm, with some being further lengthened by other comorbidities[42]. Specific demands include the length of meal times, the need to modify food and prepare separate meals from those of the rest of the family, the perceived inability of children to feed themselves, the messiness caused during meals, and the pressure of providing sufficient food of good enough quality[42]. Messiness was a particular concern; it could lead to extra chores and might deter others from helping caregivers with feeding[42]. The Brazilian study confirmed that time taken is an issue; but also the timing of the challenges in relation to other issues with which the woman has to contend [38]. Mothers' comments suggest that feeding was often physically exhausting as well as stressful, both for the mother and child [42,43].

## Health professionals

As with the CZS studies, difficulties when feeding may be attributable to physical signs of choking, vomiting and swallowing difficulties. Health professionals considered respiratory functions, specifically choking, aspiration, pneumonia and chronic lung disease, to have a large impact on the physical comfort of the child[44].

**Parents feel that the information and support provided to them by health professionals is mostly inadequate.** The inadequacy of information provided by health professionals was reported in four studies[37–40] by multiple respondents. Mothers in the UK and Australian studies reported seeking answers to their questions and uncertainties, but the advice forthcoming was inconsistent or there was 'no answer' at all, [39,40] with the result that some mothers reported not wanting to access services [39]. Participants reported a 'lack of skilled help', 'mixed messages' and a 'lack of information provided on feeding'[39]. Mothers in the same study reported not being given 'correct' advice on expressing breast milk or bottle-feeding, with the result that one mother reported attempts at bottle feeding being 'frightening'[39]. Some health professionals, when faced with feeding difficulties, were described as being 'out of their depth' and unable to provide appropriate growth charts[39]. Mothers of Down's Syndrome infants mothers reported inconsistency in the information and advice being provided [40].

Even when attention and support was provided, it was reported that it could be considered inadequate [39]. Mothers in the study from Portugal felt that health professionals did not encourage breastfeeding and did not pass accurate information, confusing mothers about practices they should have, leaving doubts, discomfort, and a demotivating environment [37]. Health professionals were often insensitive to stress and other concerns and parents reported feeling they have to seek information themselves. Mothers in the Brazilian study described how family members and friends often sought to provide support when this was not forthcoming from the health professionals[38]. Every family had resources and previous breastfeeding experience upon which to draw, with beliefs and values being passed on among family members particularly in connection with breastfeeding support[38].

Finally, mothers in the Australian study complained that attending scheduled appointments with health professionals when their child was having feeding difficulties could be stressful[40]. Support from health professionals within nutrition services was variable; when visiting nutrition clinics, some were not referred for treatment, some were given advice that they felt was inappropriate, while others were referred for treatment but had difficulties attending appointments and did not receive follow-up. This disconnect with nutrition services seemed to contribute to limited care-seeking habits of caregivers[42].

However, mothers in the UK study reported good support from neonatal unit staff, as well as support and advice on nutrition, if not on breast-feeding [39]. The Brazilian study also reported support received from health professionals, particularly nursing, maternity and those

providing home visits as well as team members from the Human Milk Bank[38]. Parents in the Australian study of infants who had been in neonatal intensive care also reported good support from staff both for them and their children, as well as in terms of making their child well enough to go home[40].

**Parents report feeling they have to seek information themselves.** In the presence of uncertainty, and the absence of adequate information and support from health professionals, mothers of infants with Downs Syndrome reported having to seek information on feeding themselves, both ante- and post-natally, including on choice of breast pumps and infant weight gain[39]. However, one mother in this UK study also reported being too exhausted or stressed to conduct her own research. Another mother reported, while feeling that the information should have been provided to her, that the process of seeking and finding information was a positive experience as it gave her a sense of control[39]. Participants from the study in Porto described how they identified other parents through Internet forums (on sites that specifically address Down's Syndrome), and Facebook groups, and were frustrated when they attended appointmets and seemed to have greater knowldege than staff [37].

**Parents report a general lack of control.** Two studies reported that mothers felt a lack of control over the feeding decisions made for their child, whether it was to persist with breast-feeding, to stop breast-feeding or to adopt a combination of breastfeeding, expressing or bottle-feeding [37,39]. These mothers reported a 'loss of power' regarding feeding, having their preferences 'overridden' by health professionals[39] and that health professionals acted as the 'gatekeepers' to services[39]. In the UK study of mothers of infants with Downs Syndrome, a mother reported being 'pushed' from breast-feeding to expressing, then from expressing and tube feeding to formula feeding[39]. This lack of choice also surfaced in the other study of Down's Syndrome (DS) from Portugal[37] where women were denied the opportunity to breastfeed their children soon after their child's birth due to factors such as their child's health problems, their child's hospitalization in a neonatal unit, or due to the opinions of health professionals who thought the practice of breastfeeding in children with DS was impossible. As a consequence, breast milk was offered with the use of a bottle, a syringe, and a nasogastric tube. Mothers in an Australian study reported feeling that their 'maternal competence' was being challenged, and their bond with their child was affected if others took control of feeding when the mother's actions were not proving successful[40]. In both the UK study and the Australian parental study, mothers reported they were frustrated at the control exercised by health professionals and within maternity hospital environments [39,40]. However, fathers in one study[40] reported that the involvement of health professionals alleviated the stress and frustration as someone else was taking control. Fathers were also positive about being able to give expressed breast milk as it gave them a feeding role that they otherwise would not have had [40].

The Brazilian study also asserted that knowing that every child is a single individual and that differences in breastfeeding occur across individuals helped to strengthen the determination of one mother in continuing to insist on breastfeeding her son with Down's Syndrome [38]. Mothers were very aware that health professionals should respond to these differences and, if necessary, update their knowledge to be able to deal with such cases[37]. The Australian study of health professionals saw them highlighting the need to recognize the individuality of families' needs[44]. These health professionals asserted that "optimal communication from health professionals should be non-aggressive, supportive, non-judgemental and respectful of the parents' viewpoint" [44]. The Australian health professionals in this study identified "provision of educational information, acknowledgement of the parent as main carer and choosing the appropriate time to communicate difficult information" as being particularly important

[44]. However, these health professionals also recognised that parents experience loss of control when dealing with the medical system[44].

In the study of neonatal feeding difficulties a mother reported that, while expressing was 'painful', it did give her a sense of 'doing something'[40]. In the study of infants with Downs Syndrome, another mother reported that her child was 'taken away' to be fed to support weight gain, while another mother, a health professionals herself, took control of the situation and refused to allow anyone else to take away her baby to be fed [39]. The need to facilitate women's infant feeding choices was also clearly expressed in the Brazilian study [38].

Mothers in the study of infants who were in neonatal intensive care reported wanting to 'take their baby home' in order to be 'in control'[40]. However, one mother in the UK Down's Syndrome study reported that she had much greater control over feeding when admitted to a children's hospital compared to when she was in the maternity hospital, while another mother in the same study reported that seeking and finding useful information gave her a sense of autonomy: 'it felt like a really important proactive step to at least try to get some control over the situation'[39]. By contrast, some fathers and mothers with other small children, in the study of infants who had been in neonatal intensive care, reported that they were 'happy to let nurses care for their infant' because of the demands of caring and feeding, and the expectation that the child would be well enough to leave the hospital more quickly [40].

**Parents report that the infant's weight gain can be the overwhelming focus both for them and for health professionals.** Parents of children who had been in neonatal intensive care reported that the priority for them, and for health professionals, was their child's weight gain rather than how he or she was fed: the reported reasons were not only to see their child thriving, but to do so with the aim of 'getting home' and out of hospital as quickly as possible [40]. Health professionals recognised that discussions regarding weight gain often revealed discordance between parents' and health professionals' wishes[44]. Feeding and weight gain, for reasons other than discharge, such as surgery, were also described by one mother in the other study [39]. In the same study, a mother reported how weight gain was prioritised over other factors in decisions made over feeding choices [39].

## Support

**Training programmes for parents.** Several studies examine training programmes, often in low-cost settings, to improve feeding of children with cerebral palsy [41,42,43,44,46]. Programmes include training on head positioning[42,43] and on mixing foods to an appropriate consistency. Bangladesh and Ghanaian training programmes for cerebral palsy aimed to alleviate stress from choking by offering specific guidance on positioning [41,42]. Table 6 summarises potential implications of this indirect evidence when planning Zika training programmes.

Training programmes offer a potential source of practical advice with caregivers feeling able to give more diverse foods after having been given advice on appropriate modifications. Caregivers in this Ghanaian study reported the need to modify the consistency of foods, making them softer or liquidized, to enable feeding[42].

Parents appreciate improved support from others as a result of sharng information that they themselves have received from training programmes. One mother of a two-year old who attended a training programme described how she was able to share knowledge on choking with her own mother which offered respite in the toddler's care [43]. This successful sharing or participation of other family members in training was also reported in another study [41]. Some mothers in the study from a NICU reported that intervention was a 'helpful' step, 'restoring positive experiences around feeding'[40]. Family and professional support helped

**Table 6. Implications of indirect evidence [41–44] for Zika parental training programmes.**

| Content | Delivery | Logistics |
|---|---|---|
| Head positioning and jaw stability | Information to share with other family caregivers | Difficulties accessing transportation |
| Adapting feeding methods: Foster self-feeding skills | Traditional pedagogy, discussion, participatory and experiential activities | Difficulties paying transport costs |
| Adapt feeding method: Use sensitive, proactive and responsive feeding methods (including hygienic cooking and feeding practices) | Use of visual aids including 20-min video drama created especially for programme | Absence of fathers |
| Mixing foods to required consistency | Connection with other caregivers of affected children | Lack of resources to buy nutritious foods |
| Alleviating stress from choking | | Low-cost seat ($5/child) made of reinforced cardboard |
| Introducing more diverse foods | | Plastic teaspoon and cup bought in local market |
| Use of appropriate utensils | | |
| Available financial welfare support | | |

them manage their different feelings and to enhance their positive feelings and thus help in the maintenance of breastfeeding, without early weaning.

Some mothers described how family members and friends often sought to provide support when not being received from the health professionals. Sharing information with parents facing similar challenges was reported to be one of the most effective ways of learning. The mutual support and sense of "family" were considered important contributions of training programmes[43]. Evidence from cerebral palsy workshops in Ghana and Bangladesh suggests that it may be difficult for mothers to attend group training provision if they are unable to access transportation or pay travel costs. The absence of fathers further contributed to a lack of resources in the Ghanaian study, with no family accessing any social protection programmes at the outset of the training [43].

**Resource considerations.**   Australian health professionals considered the parental burden of time, physical effort, financial and emotional costs of feeding to be high[44]. One mother from Ghana expresses this intersecting relationship between the physical dependency of the child, the economic implications and caregiver stress: "I am unable to leave him and even go and work because of his condition. I need to work and support him and the other children but I just cannot. This makes me very unhappy"[42]. Mothers may be unaware of available financial welfare support[43].

Poverty shaped the wellbeing of the majority of Ghanaian caregivers and their ability to implement change from the training programme[43]. The absence of fathers, who may abandon the mothers when faced by the perceived added burden of a child with cerebral palsy, could further contribute to a lack of resources[43]. This Ghanaian study found that "Fathers were almost completely absent from the household, whether because of separation or divorce, working away from home, and infrequent home visits"[43]. Adams observes, in the context of Bangladesh, that: "in situations of poverty, problems are exacerbated by factors such as lack of resources to buy nutritious food, limited time and facilities for cooking special recipes"[41]. Mothers and their children may also face lack of access to rehabilitation and health services to deal with associated complications"[41], including lack of transport [43]. Parents with better economic circumstances had additional health insurance to address the deficits of the social care services. However, some specialties were not registered with health providers thereby occasioning further expense as mothers cared for their child.

### GRADE-CERQual and summary

Given the richness, coherence and relevance of the evidence, and the low risk of bias of most studies, it is possible to have moderate confidence in almost all findings regarding feeding of infants with with severe disability or nonprogressive, chronic encephalopathies (see Table 7). The sample included eight studies, four from economically highly-developed, western countries. Further, primarily the perspectives of parents (principally mothers) were provided. Despite being the subject of a substantial amount of the data, health professionals were only included in the sample of one study[44]. As a result, while these findings on parents' values and preferences for infant feeding are extremely important, it must be cautioned that the sample of studies is localised to a particular group and context. Studies on Down's Syndrome have tended to focus on breast feeding and functional difficulties such as positioning and latching on. In contrast, the studies of cerebral palsy tend to explore challenges throughout infancy and childhood, including difficulties associated with weaning and then solid foods. The external validity of the findings, especially as they apply to Zika, is therefore limited. However, given the strengths and relevance of these indirect studies, they garner relatively higher confidence in their findings than the Zika review studies.

## Discussion

Many themes were found to be shared between the direct and indirect evidence, principally around parental anxieties and stresses, the need for information and advice, and the difficulties arising from a lack of financial means (Box 1).

Findings from the indirect evidence studies of infants with severe disability and nonprogressive, chronic encephalopathy do resonate with those in the CZS-specific literature, in particular, the stress and anxiety felt by mothers in particular, as a result of difficulties with feeding; and uncertainties about how best to proceed with feeding a baby of infant. It is clear that breast-feeding represented the preference for all mothers in these studies in both reviews, and the inability to do so adversely affected parents' bonding with their child, and rendered feeding a more time-consuming challenge. Parents also expressed preferences for bespoke training to help them manage their child's condition, and financial support and resources, where required.

The key area of difference between the two sets of evidence concerned the information and support specifically provided by health professionals, and in particular the implied need for more individualised care and shared-decision-making. Parents of these children therefore reported preferences for greater information and support from health professionals. The quality and quantity of findings on this theme from the non-CZS studies (the indirect evidence) offered a stark contrast to the respective CZS evidence. This doubtless relates in part to socio-economic differences by geographical location: the six CZS-specific studies were conducted in north-east Brazil, while four of the eight 'indirect' evidence studies were conducted in economically developed countries: UK, Australia (n = 2) and Portugal.

Another key difference, explainable given different population samples, is whether mothers are able to find their own information. Paradoxically, mothers from developed countries expressed a feeling of lack of control, perhaps indicating higher expectation and consumer-orientation from health services. Another key difference between the 'direct' (CZS studies) and 'indirect' (infants with severe disability and nonprogressive, chronic encephalopathy) evidence was the frequently expressed upset and concern, specifically around choking and potential suffication, for children with CZS, with the possible related need for First aid training. Evidence from low resource contexts suggests that fears of choking are also ameliorated

**Table 7. Non-CZS review—GRADE-CERQual summary of qualitative findings for feeding in Infants with severe disability or nonprogressive, chronic encephalopathies.**

| Summary of review finding | Studies contributing to the review finding | GRADE-CERQual assessment of confidence in the evidence | Explanation of GRADE-CERQual assessment |
|---|---|---|---|
| Parents report that they often do not know how best to feed their child | [39,40] | Moderate confidence | Two studies with minor concerns about coherence and methodological limitations. The data are rich, but there are serious concerns about adequacy (only two studies) and relevance (only from high- resource settings) |
| Parents report that feeding an infant who has difficulty feeding can be time-consuming and demanding | [38–40,42,44] | Moderate confidence | Five studies with minor or moderate concerns about coherence and methodological limitations: four studies are of high quality and the data from all studies are rich. Minor concerns about adequacy because there are five studies, and about relevance because these studies were with two exceptions (Ghana, Brazil) conducted in high income settings (UK, Australia). |
| Parents experience frustration, stress and bonding concerns when feeding their child is problematic | [38–40,42,43,44] | Moderate confidence | Six studies with minor concerns about coherence, and minor or moderate methodological limitations. The data are rich. Minor concerns about adequacy with six studies, and about relevance because studies were conducted across diverse socio-economic settings. |
| Parents feel that the information provided to them by health professionals is mostly inadequate; Parents feel that the support provided to them by health professionals is mostly inadequate | [37–40,42] | Moderate confidence | Five studies with minor concerns about coherence and minor or moderate methodological limitations: the studies are high quality and the data are rich. Minor concerns about adequacy with five studies, and moderate concerns about relevance because these studies were mostly conducted in a particular socio-economic context (UK, Australia, Portugal, Ghana). |
| Parents report feeling they have to seek information themselves | [37,39] | Low confidence | Two studies with minor concerns about coherence; and minor or moderate methodological limitations: the studies are high quality and the data are rich. There are moderate concerns about adequacy and relevance because there are only two studies from limited settings. |
| Parents report a general lack of control | [37–40] | Moderate confidence | Four studies with minor concerns about coherence and minor or moderate methodological limitations: the studies are high quality and the data are rich. Moderate concerns about adequacy because there is only five studies, but minor concerns about relevance because they are from across multiple settings (Australia, Brazil, UK, Portugal). |
| Infant's weight gain can be the overwhelming focus both for them and for health professionals | [39,40,44] | Moderate confidence | Three studies with minor concerns about coherence and methodological limitations: the studies are mainly high quality and the data are rich. Moderate concerns about adequacy because there are three studies, and about relevance because these studies were conducted in a particular socio-economic context (UK, Australia). |
| Training can alleviate concerns with choking and positioning may avoid risk of vomiting | [40–44] | Moderate confidence | Five studies with minor concerns about coherence, and minor or moderate methodological limitations. The data are rich. Minor concerns about adequacy with seven studies, and about relevance because studies were conducted across diverse socio-economic settings. |
| In situations of poverty, feeding problems are exacerbated by lack of resources to buy nutritious food, limited time and facilities for cooking special recipes and lack of access to rehabilitation and health services. Mothers may lack welfare financial assistance or support from the fathers. | [41–44] | Moderate confidence | Four studies with minor concerns about coherence and minor or moderate methodological limitations: three studies are high quality and the data in all studies are rich. Moderate concerns about adequacy given there are only four studies. Relevance has only minor concerns with three studies from a low-resource socio-economic context |

The full GRADE-CERQual Evidence profile table is available in S6 Table: GRADE-CERQual Evidence profile: Feeding in infants with severe disability or nonprogressive, chronic encephalopathies.

## Box 1. Key Findings

The key findings might be summarised as follows:

### Parental anxieties and stresses

- Parents report general uncertainty about infant feeding options; they often do not know how best to feed their child because the child frequently chokes or has difficulty swallowing (CZS [low confidence in findings] and Non-CZS [moderate confidence in findings]);

- Even if parents possess information on how to manage feeding difficulties, they fear that they might be doing or do something wrong, and that they might suffocate the child (CZS [moderate confidence in findings]);

- Parents and others report that feeding a child with swallowing difficulties makes them stressed and anxious (CZS [very low confidence in findings] and Non-CZS [moderate confidence in findings]);

- Parents experience frustration, stress and bonding concerns when feeding their child is problematic (CZS [very low confidence in findings] and non-CZS [moderate confidence in findings]);

- Parents report that the burden of feeding can be time-consuming (CZS [very low confidence in findings] and Non-CZS [moderate confidence in findings])

- Mothers report that the burden of feeding can be stressful and demanding (CZS [very low confidence in findings] and Non-CZS [moderate confidence in findings])

- Mothers report that the burden of feeding falls on them (CZS [very low confidence in findings])

### Health Professionals

- Parents feel that the information provided to them by health professionals is mostly inadequate (CZS [very low confidence in findings] and Non-CZS [moderate confidence in findings]);

- Parents feel that the support provided to them by health professionals is mostly inadequate (Non- CZS [moderate confidence in findings]);

- Parents report feeling they have to seek information themselves (Non-CZS [low confidence in findings]);

- Parents report feeling a general lack of control (Non-CZS [moderate confidence in findings]);

- Parents report that the infant's weight gain can be the overwhelming focus both for them and for health professionals (Non-CZS [moderate confidence in findings])

### Support

- Parental anxieties can be alleviated by Training programmes (CZS [very low confidence in findings] and Non-CZS [moderate confidence in findings]);

- Parents decisions can be affected by Resource considerations (CZS [very low confidence in findings] and Non-CZS [moderate confidence in findings]);

- Feeding difficulties may be exacerbated in situations of poverty, particularly given absence of financial support from the state or the husband (Non-CZS [moderate confidence in findings]) and because of the need to buy special food (CZS very low confidence in findings])

by training interventions in infants with severe disability and nonprogressive, chronic encephalopathies.

Concerns among parents about financial costs and the costs of accessing special foods are shared between the CZS and non-CZS studies conducted in resource-limited settings (Bangladesh, Ghana). Finally, the impact of feeding difficulties on the maternal-infant bond figures prominently in the literature for infants with severe disability or nonprogressive, chronic encephalopathies, and is present in the CZS evidence but mentioned in only one study[35].

Research evidence is lacking on the efficacy of providing primary caregivers of infants affected by zika-related complications with additional support such as social support, respite care, medical support, caregiving competence support, and financial support and their relationships to infant feeding complications, morbidity, mortality, growth and development outcomes. As infants and children affected by CZS mature, there is an associated need to design appropriate care recommendations and support programs. A curriculum delivered to 70 mothers of children with CZS-associated abnormalities, initiated conversations with the mothers about healthy eating, the introduction of food and swallowing disorders. Mothers in this study who enjoyed better economic circumstances obtained health insurance for their children to address social care service deficits. However, some specialties were not registered with health providers thereby occasioning further expense as mothers cared for their child[35].

A review discussing considerations for family support and services in case of children with CZS[47] suggested the following family implications: (1) the severity of the impact on children with obvious abnormalities at birth, coupled with the anticipation of a lifetime of caregiving and economic burdens; (2) uncertainty about the unfolding consequences, both for obviously affected children and for exposed children with no symptoms at birth; (3) a lack of specialized professional knowledge about the course of the disease or treatment options; and (4) social isolation, a lack of social or community supports, and potential stigma. This implies that affected families require extensive support for health care services and other type of services as well as information and surveillance.

The impact of having a child with neurodevelopmental abnormalities, associated with CZS, adds complexity and challenges to an already challenging life event. Caregivers of CZS-affected infants face the challenges of a complex day-to-day routine, often exacerbated by a need to give up their employment, by the extra costs of modifications and special equipment, and by a lack of family or community support.

Indirect evidence focused on infants with Down's syndrome and cerebral palsy; the former focusing on breast-feeding and the latter on other types of feeding. A focus on these two conditions resulted in a synthesis of qualitative studies of infants with severe disability and nonprogressive, chronic encephalopathy that was distinct from the synthesis of CZS studies, which focused on diverse feeding approaches and types of food. Also, in comparison to the synthesis

of CZS studies, most of the included research was high quality. Data were collected using different methods and included observation of parents and family, as well as approaches that directly elicited their values and preferences. One rich study explored the views of diverse health professionals from Australia.

Current management recommendations for dysphagia include postural correction, thickened feeds, pacing and spoon placement technique. Recommendations for avoiding gastro-esophageal reflux include slow feeding, keeping the infant upright after meals and elevating the head of the bed with support under the mattress. Proper positioning is also an important modification for managing a child with abnormal tone. Additional strategies for children with neurologically involved feeding and swallowing problems include adjusting the child's environment (increasing or reducing stimuli depending if the child has hypotonicity or hypertonicity), sensory stimulation (direct stimulation of the lips and oral cavity and modification of foods and liquids regarding texture, volume, temperature, and taste) and using adaptive equipment (one-way, slit-valve nipples, nosey cups, and Maroon Spoons)[48].

While most training studies did not conform to the inclusion criteria for the population age group, or for qualitative research, they could form a further basis for designing specific training programmes for children with CZS-associated feeding difficulties. Even if training programmes may not reduce the duration of feeding they may help to equip parents with realistic expectations about the time to be taken. Group training not only provides a forum for providing formal information but also an additional resource of mutual self-help. Equally importantly, however is the role of such groups in providing emotional support and in helping to cultivate realistic expectations of normative daily routine. Furthermore, it was not simply the attendees themselves who benefited from such provision; there was evidence of cascading information and practical advice to others increasing the capacity of informal support from relatives, for example. Conversely, deficiencies in the care, support and respect encountered by caregivers of infants in general were exacerbated or aggravated under the extra load imposed by the impact of the Zika virus on routine infant feeding.

## Strengths and limitations of this review

To our knowledge, this is the first meta-synthesis of the implications of Zika virus for infant feeding. Epidemiological evidence suggests little cause for concern in relation to transmission of Zika virus via breast milk. The consequences of Zika infection in either an infant or a breast feeding mother appear negligible. No data are available to establish how the risk of Zika infection through breast feeding compares with the risk through direct mosquito bite. However, the effect of neurodevelopmental abnormalities associated with in utero infection on infant feeding is potentially profound with consequences that may extend into adult life.

Systematic reviews inevitably depend on the richness and quality of data from primary studies that have already been collected and reported. In reviews of qualitative studies, these data have already been interpreted through the lens of what is seen to be important by the primary authors. A concentration of studies from a specific region (as with six Zika studies from Brazil) strengthens the relevance of findings from that vicinity but may not increase confidence in transferability of those findings to other settings, thus occasioning a role for the use of indirect evidence[18].

Use of indirect evidence invokes some rationale for comparability between source studies and target condition[18]. Practical difficulties of feeding an infant with neurodevelopmental challenges may be very relevant to difficulties experienced by Zika caregivers, especially where studies derive from a low resource setting. Although the original intent was to only include studies that reported on the views, attitudes and values associated with the Zika virus, the

WHO guidelines development group argued convincingly for the potential value of a wider indirect evidence base, populated by studies of cerebral palsy and Downs Syndrome. The inclusion of eight additional studies of neurodevelopmental disorders offers a holistic view of the phenomenon of interest, not achievable by the limited six CZS-specific studies. They broaden values and preferences to other regions of the world, while resonating with the specific CZS experience. It should also be noted that their addition to the current evidence is important because the indirect evidence is relatively more robust methodologically, the data were richer, more relevant, and the settings were broader, so external validity was greater than for the direct evidence. The confidence in the findings of the former was therefore also greater than for the direct evidence of the CZS sample of studies.

The use of translation software at the inclusion stage of the review extended the evidence base beyond English language studies; particularly important given the regional distribution of the Zika virus. Translation facilities (particularly in Portuguese) were sufficiently accurate to allow determination of inclusion and exclusion, to permit classification and to add to the themes supported by other, English language, studies. A further feature was the requirement to keep updating the searches to identify new incident studies right up to the submission of the manuscript–given this most volatile of qualitative evidence synthesis topics. This is attested to by the fact that all CZS studies derived from publications published between 2017 and 2019. The indirect evidence review enabled the broadening of findings beyond the initial South American focus. GRADE-CERQual assessments indicated that confidence in the CZS findings was only low or very low, but in the related non-CZS findings, it was moderate or high, reflecting the quantity and quality of these indirectly relevant non-CZS studies. The inclusion of the wider range of settings, viewpoints, and study types, once the indirect studies were added, thus benefitted the overall review and the understanding of the phenomenon of interest.

A further line of inquiry might include values and preferences with regard to feeding tubes [49]. Literature searches for studies of infants with severe disability and nonprogressive, chronic encephalopathy serendipitously retrieved articles on feeding tubes but this had not been prespecified as a focus. Further targeted information retrieval might retrieve qualitative studies that make a separate qualitative synthesis viable. However, it is not known what proportion of affected infants might require this intervention or, indeed how available this option is in low-resource settings, and, therefore, the relevance of this literature to the overall review is uncertain.

However, a key finding has been the need for research that focuses on feeding challenges within this particular patient group, and explores the approaches and support that parents find most effective for helping them to overcome and manage the feeding difficulties presented by an infant with CZS. Such work would provide vital information for infant feeding guidelines.

## Conclusions

This review demonstrates that practical and economic concerns around having a child with CZS-related abnormalities, or indeed other neurodevelopmental disorders that impair infant feeding in a similar way, add to an already stressful life event. Stress, anxiety and uncertainty are common experiences. These anxieties may be exacerbated by limitations around information provision and counselling, particularly in low- and middle- income resource settings. However, indirect evidence, from mothers of infants with cerebral palsy, suggests that group workshops for women with affected infants are considered useful and may be address deficiencies in the wider health system. Whether information provision and counselling is best delivered in a specific context, tailored to the condition, integrated within specific provision for children with neurodevelopmental disorders, or delivered within routine infant feeding provision remains to be resolved. Regardless of the exact mode of delivery it is clear that mothers

and infants require information and emotional support if they are to receive infant feeding provision that they consider to be responsive to their values, beliefs, and needs.

## Supporting information

**S1 Fig. CASP quality assessments for studies on values and preferences of Congenital Zika Syndrome (CZS) populations.**
(TIF)

**S2 Fig. CASP quality assessments for studies on infants with severe disability or other non-progressive, chronic encephalopathies.**
(TIF)

**S1 Table. STARLITE specification of each search method.**
(DOCX)

**S2 Table. Principal search strategy.**
(DOCX)

**S3 Table. Themes and illustrative quotations: Feeding infants with Congenital Zika Syndrome.**
(DOCX)

**S4 Table. GRADE-CERQual evidence profile: Feeding infants with Congenital Zika Syndrome.**
(DOCX)

**S5 Table. Themes and illustrative quotations: Feeding in infants with severe disability or nonprogressive, chronic encephalopathies.**
(DOCX)

**S6 Table. GRADE-CERQual evidence profile: Feeding in infants with severe disability or nonprogressive, chronic encephalopathies.**
(DOCX)

## Author Contributions

**Conceptualization:** Christopher Carroll, Andrew Booth.

**Data Curation:** Andrew Booth.

**Formal analysis:** Christopher Carroll, Andrew Booth, Fiona Campbell.

**Funding acquisition:** Christopher Carroll, Andrew Booth, Clare Relton.

**Investigation:** Christopher Carroll.

**Methodology:** Christopher Carroll, Andrew Booth.

**Writing – original draft:** Christopher Carroll, Andrew Booth.

**Writing – review & editing:** Christopher Carroll, Andrew Booth, Fiona Campbell, Clare Relton.

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
