## [Decision Letter · Decision Letter 0]

15 Jul 2020

Dear Dr. Carroll,

Thank you very much for submitting your manuscript "What are the implications of Zika Virus for infant feeding? A qualitative evidence synthesis" for consideration at PLOS Neglected Tropical Diseases. As with all papers reviewed by the journal, your manuscript was reviewed by members of the editorial board and by several independent reviewers. The reviewers appreciated the attention to an important topic. Based on the reviews, we are likely to accept this manuscript for publication, providing that you modify the manuscript according to the review recommendations. 

Thank you again for your submission to our journal. We hope that our editorial process has been constructive so far, and we welcome your feedback at any time. Please don’t hesitate to contact us if you have any questions or comments.

Sincerely,

Alberto Novaes Ramos Jr, M.D., M.P.H., Ph.D.

Guest Editor

Louis Lambrechts

Deputy Editor

Reviewer’s Responses to Questions

**Key Review Criteria Required for Acceptance?**

**Methods**

-Are the objectives of the study clearly articulated with a clear testable hypothesis stated?

-Is the study design appropriate to address the stated objectives?

-Is the population clearly described and appropriate for the hypothesis being tested?

-Is the sample size sufficient to ensure adequate power to address the hypothesis being tested?

-Were correct statistical analysis used to support conclusions?

-Are there concerns about ethical or regulatory requirements being met?

Reviewer #1: This novel paper is a WHO-commissioned qualitative review of mothers’ values and preferences regarding feeding of infants with congenital Zika syndrome. The rationale for the study, objectives, study design including search strings, study selection, and data extraction were described thoroughly and were appropriate for an ethnographic, qualitative study. The rationale for broadening the review to include mothers of infants with conditions that confer similar feeding difficulties was very helpful, and as outlined on p. 58 lines 7-9, added significantly to the applicability and importance of the findings.

Reviewer #2: The method used to evaluate qualitative studies published on Zika and infant feeding meets the newspaper’s recommendations. The objective is clear and directly related to the methodology used and the exposure of the results. 

Suggestion: make it clear which criteria for selecting the article was used, as the number of publications on the topic is high, but only six articles met all the criteria. 

Make it clear if the authors read the abstracts of all articles selected in the 1st phase. Many authors do not use the keyword qualitative research in publications and this fact can exclude many articles that have been published. I suggest the authors make it clearer how this analysis of the articles found in the initial review was made.

Reviewer #3: (No Response)

**Results**

-Does the analysis presented match the analysis plan?

-Are the results clearly and completely presented?

-Are the figures (Tables, Images) of sufficient quality for clarity?

Reviewer #1: With many qualitative studies, the described extraction words, phrases, and themes can seem broad, but this study managed to capture some very specific commonalities and presented them in a cogent manner that sets the stage for proposed further study.

Reviewer #2: There was a good description of the analysis of the selected articles and the summary tables have a good clarity and description of the results.

Table 6 brought a certain difficulty in the reviewer’s understanding: why were two studies with children with down syndrome included? If these studies do not bring results from children with Zika virus, I do not consider it important to have this picture mentioned.

Reviewer #3: (No Response)

**Conclusions**

-Are the conclusions supported by the data presented?

-Are the limitations of analysis clearly described?

-Do the authors discuss how these data can be helpful to advance our understanding of the topic under study?

-Is public health relevance addressed?

Reviewer #1: The description of strengths and limitations starting on p 57 line 5 is very thorough and clearly carefully thought out. Again, I appreciated the suggestion that inclusion of mothers of infants with Downs syndrome and CP may actually enhance our understanding of the difficulties they face. The one consideration I would suggest is further development of the challenge of the geographic confines of Zika. As alluded to on p. 26, line 19, it begs to reason that the majority of studies were out of Brazil, as Brazil has some of the highest breastfeeding compliance and some of the highest prevalence of Zika virus, and while the indirect evidence broadens applicability, it also introduces a host of potential socioeconomic and cultural confounders. Specific mention of social support and economic hardship as contributors to infant feeding difficulties in Ghana is described, but I suspect there are a multitude of factors that may have arisen that would add to our understanding of the hardships parents of infants with congenital abnormalities that affect feeding experience-- perhaps even a short table of some of the less notable but interesting findings?

Reviewer #2: The study brought some conclusions from the experience and perception of women who have children with microcephaly. I studied the nutrition of children with microcephaly in a state in the Northeast (with a high prevalence of births of children with microcephaly, due to zika virus) and I ask the authors: in none of these studies did women talk about the use of bottles to avoid choking on the child? 

The authors must reinforce that the government must adopt income transfer programs to support these families with children with microcephaly. In addition, I consider it important for the authors to mention the importance of multidisciplinary teams in the follow-up of children with difficulties in food intake.

Reviewer #3: (No Response)

**Editorial and Data Presentation Modifications?**

Reviewer #1: Minor spelling/punctuation errors, eg p 3 line 13 "celebral," and p. 7 line 7 eliminate the "?"

Reviewer #2: Minor revision

Reviewer #3: (No Response)

**Summary and General Comments**

Reviewer #1: This study describes a significant issue in social support and provider education for mothers of infants with CZV or other congenital feeding challenges. Provider education in supporting breastfeeding alone is lacking in many healthcare systems, and this is a well-written charge to improve provider understanding, dissemination of information, and need for extra support for these families.

Reviewer #2: The study is relevant and brings to reflection a theme that is little discussed and published in international journals. Qualitative studies can show many conditions of difficulties experienced by mothers with children with microcephaly. The analysis was well done, bringing only the doubt if there was a reading of all abstracts of the articles identified in the databases. Qualitative studies are important to be published, as they bring a more in-depth analysis and can offer services ideas on how to deal with these problems faced by mothers of children with microcephaly.

Reviewer #3: Review for manuscript PNTD-D-20-00781: What are the implications of Zika Virus for infant feeding? A qualitative evidence synthesis

I. Summary: 

This is generally a well-written manuscript describing a study that was developed to synthesize the available qualitative evidence related to the values and preferences of infant feeding options from caregivers, healthcare practitioners, and policymakers to feed children with congenital Zika syndrome (CZS) and other nonprogressive chronic encephalopathies. 

The main goals of this review are to: 

(1) Synthesize qualitative evidence related to the fears, perceptions, experiences, beliefs values, and preferences of feeding options from caregivers, healthcare practitioners, and policymakers to breastfeeding children with CZS and similar non-related disorders (e.g., encephalopathies)

(2) Identify the preference of infant feeding options (breastfeeding either from the breast or expressed, formula feeding and mixed feeding) in children with conditions that result in feeding difficulties

(3) Inform infant feeding guidelines

Corresponding to aim1, the authors successfully identified evidence specific to the values of caregivers when attending children with congenital feeding difficulties. They describe several challenges regarding the uncertainty from caregivers of how and what feeding practices are optimal to feed a child experiencing feeding difficulties. There was also a sense of infant feeding being burdensome for it might be time-consuming and potentially hazardous to the infant and thus, there is a degree of stress and anxiety when feeding. The authors also report on social stigma and economic strains as leading factors preventing caregivers to bond with children with Zika related and other non-Zika neuropathies. 

Regarding healthcare professionals the overall report suggests that the lack of proper training to manage children with feeding difficulties is a leading reason for the parents receiving mixed and often not adequate instructions for feeding their children. Also, it was reported that the caregivers often look for information themselves because of the limited information and support offered by their healthcare providers. These conclusions seemed to be consistent in both CZS and non-CZS groups.

Concerning aim 2, findings of feeding preferences are not described in the results section. Perhaps, there was limited or no evidence available regarding the preference of feeding practices. If available, this narrative could be improved by including a statement describing preferred feeding practices among caregivers and if not available, including a brief statement that the evidence was not present might be useful to the reader. 

On aim 3, the authors highlight the limited access to tailored programs and lack of support from governmental agencies for both parents and healthcare practitioners, which is useful knowledge for informing future infant feeding guidelines in the context of feeding difficulties due to biological reasons. However, the results section did not mention if the studies also reported on training healthcare providers, or perhaps it was not found within the reports. If that’s the case it would be useful to include a brief statement of the lack of evidence in the hopes of promoting further research to fulfill these gaps. 

II. Paper Strengths and contributions to the field:

Parents, caregivers, and families of children born with congenital Zika disorders and other nonprogressive chronic encephalopathies often struggle to feed these infants. This review offers a compilation of perspectives from the caregivers to inform healthcare practitioners and policymakers of their needs to support optimal infant feeding among these populations. Identifying these feeding preferences is relevant to improve management and feeding recommendations among infant populations suffering from biological conditions that lead to feeding difficulties.

III. Concerns: 

The title of the review can be improved as the paper has a broader scope. The scope of this study is not limited to synthesize evidence from studies that include infants with CZD (6 studies), instead, it also includes infants with other severe disability and nonprogressive chronic encephalopathies (8 studies).

Systematic searches for ZIKA and non-ZIKA chronic encephalopathies were performed in different but partially overlapping databases. This might limit the scope of the review. 

One of the focus questions that the study sought to address was to identify the preferences of infant feeding practices of caregivers in both the CZS and non-CZS study groups; however, these findings are not described within the main text. 

IV. Additional comments following the structure of the paper

Abstract: 

• The narrative in the result section, within the abstract, has a very narrow focus of certainty of the available evidence. While this information is essential to draft out infant feeding recommendations, the abstract could emphasize other important findings of the study.

• Conclusions could also be improved by briefly stating pieces of information that this review brought to light which are still missing from current infant feeding guidelines when children have feeding difficulties.

Introduction:

• It might be useful to include the author’s definition of what feeding support is or what it should be. 

Methods: 

• The methods are appropriate for a systematic review. However, the systematic search for CZS articles and other related chronic encephalopathies was performed in different databases and there is no clear explanation for this. Also, it is not clear why the qualitative evidence within quantitative studies was excluded.

Results: 

• It is not clear whether the authors did not find evidence regarding infant feeding preferences or was not included in the result section. 

• The authors describe detailed evidence on parental training but it remains unclear whether training of healthcare professionals was also found. This might be relevant to establish the gaps for future studies and the development of training programs that address these issues. 

Discussion:

• In box 1, the author summarizes key findings of this study. It may be more intuitively understood by the reader if the authors replace this with visual schematics and also include the certainty of the discussed evidence, which is not the same among CZS and non-CZS groups.

Other materials:

Table 1

• While the description of table 1 is “Conditions with potentially relevant infant feeding difficulties comparable to CZS” there is no direct comparison but a list. It might be more useful to the reader to include two additional columns - one that describes the condition and another that compares it with CZS. 

V. Other minor issues:

• Inconsistent usage of abbreviations through the text. Specifically, CZS, WHO, and QES.

• Redundancy of text, especially in the introduction.

VI. Reviewer’s conclusion: 

Compared to the goals that the author set out, this review focuses mainly on synthesizing the perceptions of caregivers from children with CZS and all other severe disabilities and chronic encephalopathies (aim 1). Throughout the text, the authors splits studies CZS and non-CZS but related conditions. But while the origin of these conditions might different, they both lead to feeding difficulties in infancy and thus, similar values and perceptions from caregivers may be expected. Noteworthy is that the author briefly discusses current feeding recommendations and sets out the premise of investigating the feeding preferences among caregivers of children with feeding difficulties but later doesn’t discuss if any of these recommendations were implemented by caregivers and, if so, which ones were deemed to be successful. 

Evidently, there is a need to critically evaluate the perceptions of caregivers tending children with feeding difficulties in order to develop tailored resources that better equip both parents and healthcare practitioners to manage infants with such conditions. This manuscript sets a foot in the right direction. However, in terms of improving feeding guidelines, the manuscript is limited and could be improved to serve as a resource to define gaps in the literature.

PLOS authors have the option to publish the peer review history of their article (what does this mean?). If published, this will include your full peer review and any attached files.

Reviewer #1: Yes: Margaret Dow, MD

Reviewer #2: No

Reviewer #3: No

Figure Files:

Data Requirements:

Please note that, as a condition of publication, PLOS’ data policy requires that you make available all data used to draw the conclusions outlined in your manuscript. Data must be deposited in an appropriate repository, included within the body of the manuscript, or uploaded as supporting information. This includes all numerical values that were used to generate graphs, histograms etc.. For an example see here: http://www.plosbiology.org/article/info%3Adoi%2F10.1371%2Fjournal.pbio.1001908#s5.
---

## [Decision Letter · Decision Letter 1]

18 Aug 2020

Dear Dr. Carroll,

We are pleased to inform you that your manuscript 'What are the implications of Zika Virus for infant feeding? A synthesis of qualitative evidence concerning Congenital Zika Syndrome (CZS) and comparable conditions' has been provisionally accepted for publication in PLOS Neglected Tropical Diseases.

Best regards,

Alberto Novaes Ramos Jr, M.D., M.P.H., Ph.D.

Guest Editor

Louis Lambrechts

Deputy Editor

Reviewer's Responses to Questions

**Key Review Criteria Required for Acceptance?**

**Methods**

-Are the objectives of the study clearly articulated with a clear testable hypothesis stated?

-Is the study design appropriate to address the stated objectives?

-Is the population clearly described and appropriate for the hypothesis being tested?

-Is the sample size sufficient to ensure adequate power to address the hypothesis being tested?

-Were correct statistical analysis used to support conclusions?

-Are there concerns about ethical or regulatory requirements being met?

Reviewer #1: This paper is a revision of a qualitative evidence synthesis regarding infant breastfeeding in the setting of congenital Zika syndrome. The analysis includes both direct evidence of all stakeholder experiences in breastfeeding infants with CZS and indirect evidence from experiences with infants with other chronic encephalopathies. Objectives are clearly stated with an appropriate study design. Revisions include significant detail in the review and analysis, as well as a more substantive explanation of the indirect evidence propriety. Qualitative analysis followed best practices and was appropriate for this study. There were no regulatory or ethical concerns, as this is a qualitative analysis of stakeholder experiences and perceptions.

The edits provided have substantially improved the clarity and findings by focusing and restructuring the data presentation.

Reviewer #3: (No Response)

**Results**

-Does the analysis presented match the analysis plan?

-Are the results clearly and completely presented?

-Are the figures (Tables, Images) of sufficient quality for clarity?

Reviewer #1: Analysis matches the plan. The addition and editing of several tables makes this highly readable and allows the reader to grasp the major tenets quickly— specifically, tables 4 and 7 are easily digestible and well organized. Visual quality is of sufficient quality.

Reviewer #3: (No Response)

**Conclusions**

-Are the conclusions supported by the data presented?

-Are the limitations of analysis clearly described?

-Do the authors discuss how these data can be helpful to advance our understanding of the topic under study?

-Is public health relevance addressed?

Reviewer #1: The authors have taken a “niche” topic with a qualitative analysis (often undervalued) and created a study with rigor and clarity that will contribute to healthcare provider understanding of breastfeeding issues in CZS and pave the way for additional work in an important area. Furthermore, the revisions have drastically improved both readability and applicability of their work.

Reviewer #3: (No Response)

**Editorial and Data Presentation Modifications?**

Reviewer #1: Minor punctuation errors, such as extra commas and double periods that do not affect readability.

Reviewer #3: (No Response)

**Summary and General Comments**

Reviewer #1: An important part of these revisions is the tightening of the conclusions, which counterintuitively improves how the paper paves the way for future research in this area. While CZS is thankfully decreasing in incidence, novel causes of neonatal encephalopathies and feeding difficulties will present, and this paper underscores several important public health needs in this population.

Reviewer #3: (No Response)

PLOS authors have the option to publish the peer review history of their article (what does this mean?). If published, this will include your full peer review and any attached files.

Reviewer #1: **Yes: **Margaret Dow

Reviewer #3: No

---

## [Editor Report · Acceptance letter]

29 Sep 2020

Dear Dr. Carroll,

We are delighted to inform you that your manuscript, "What are the implications of Zika Virus for infant feeding? A synthesis of qualitative evidence concerning Congenital Zika Syndrome (CZS) and comparable conditions," has been formally accepted for publication in PLOS Neglected Tropical Diseases.

Soon after your final files are uploaded, the early version of your manuscript will be published online unless you opted out of this process. The date of the early version will be your article’s publication date. The final article will be published to the same URL, and all versions of the paper will be accessible to readers.

Best regards,

Shaden Kamhawi

co-Editor-in-Chief

Paul Brindley

co-Editor-in-Chief
